# Model Recycling Framework for Multi-Source Data-Free Supervised Transfer Learning

## Abstract

Increasing concerns for data privacy and other difficulties associated with retrieving source data for model training have created the need for *source-free transfer learning*, in which one only has access to pre-trained models instead of data from the original source domains. This setting introduces many challenges, as many existing transfer learning methods typically rely on access to source data, which limits their direct applicability to scenarios where source data is unavailable. Further, practical concerns make it more difficult, for instance efficiently selecting models for transfer without information on source data, and transferring without full access to the source models. So motivated, we propose a model recycling framework for parameter-efficient training of models that identifies subsets of related source models to reuse in both white-box and black-box settings. Consequently, our framework makes it possible for Model as a Service (MaaS) providers to build libraries of efficient pre-trained models, thus creating an opportunity for multi-source data-free supervised transfer learning.

## 1 Introduction

Many existing methods for transfer learning and domain adaptation heavily rely on access to data from the source domains (Pan et al., 2010; Herath et al., 2017; Kulis et al., 2011; Zhuang et al., 2019; Yao & Doretto, 2010; Sun & Saenko, 2015; Jiang & Zhai, 2007). This situation can give rise to privacy concerns, as organizations may not want to share sensitive information; for instance, healthcare providers may be reluctant to share patient information and security system maintainers may not want to risk sharing facial recognition data for system performance updates. Additionally, there may be issues with obtaining the source data such as when it is hard to retrieve due to technical difficulties or intellectual property restrictions (Li et al., 2020b; Chen et al., 2021; Liang et al., 2020; Ahmed et al., 2021b).

Recent advancements in source-free unsupervised domain adaptation (SFUDA) have presented solutions for a scenario where source data is not accessible (Fang et al., 2022). Purposely, SFUDA utilizes pre-trained source models to improve the generalization of a model on an unlabeled target dataset. Our work is similar to other approaches in the field of SFUDA (Li et al., 2020b; Chen et al., 2021; Liang et al., 2020; Ahmed et al., 2021b), in that it addresses the practical scenario where source data is not available during training. Importantly, a crucial aspect is often overlooked by the majority of SFUDA studies. When it is assumed that source data is not accessible, then it cannot be guaranteed that the available source models have been trained on domains related to the target task. And yet, most of the works only have experimented on classic domain adaptation benchmarks, which are somewhat related by design, *e.g.*, `Digits-Five` (Peng et al., 2019), `Office-31` (Saenko et al., 2010), and `Office-Home` (Venkateswara et al., 2017), *i.e,*, domains that share the same labels but are dissimilar in feature (and ambient) space.

Our approach is unique in that we consider such a *source-free supervised transfer learning* (SFSTL) setting (Lee et al., 2019), where we do not assume source models are trained on tasks with similar feature spaces or labels. Instead, we not only experiment on classic domain adaptation datasets, but also discuss the scenario where source and target tasks have very different datasets or their label sets are partially shared. Another distinguishing factor of our work is that we consider a framework that can search a pool of pre-trained models for those that are most helpful in enhancing the performance of new tasks. Furthermore, it also

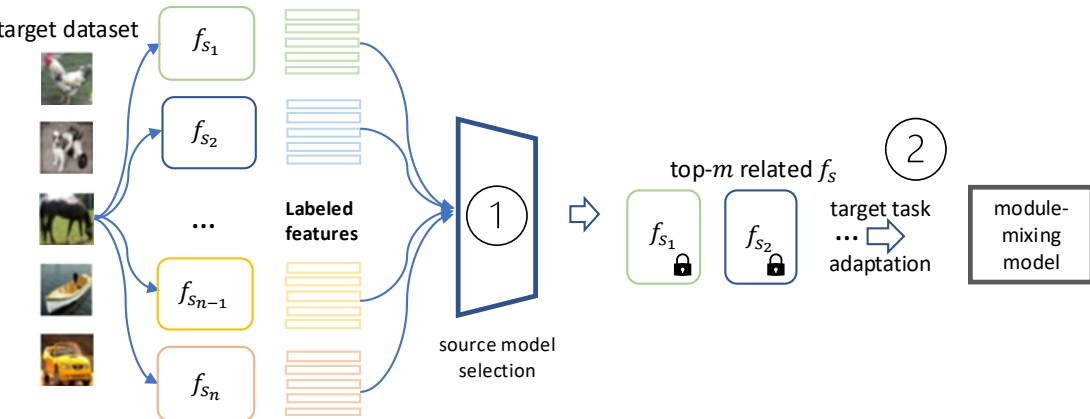

Figure 1: Proposed method in a nutshell: ① For the source model selection phase, we extract features for the target dataset with all the source feature extractors $\{\boldsymbol{f}_{s_n}\}_{n \in S_m}$, and use these features to select the top-$m$ related source feature extractors. ② For the target task adaptation phase, the selected feature extractors are used to build a module-mixing model in either white-box or black-box settings, the details of which are shown in Figure 2.

allows one to do transfer with either white-box or black-box assumptions for the source models. For the *white-box scenario*, we assume many pre-trained source models are already available on a server or service, which defines in advance the model architecture to be used by all models. In the *black-box scenario*, only the extracted features before the classification head are accessible, as large pre-trained models are considered intellectual properties nowadays, thus model details are sometimes unavailable. In either case, users can tweak parameters, such as the number of source models, learning rate, and weight decay, to obtain a better model relative to training an independent model with their own data. This process would presumably require little technical knowledge. Moreover, since many MaaS platforms would accumulate pre-trained source models over time, we also provide a parameter-efficient solution for training source models to control memory requirements.

Our contributions are summarized below: *i)* We study an under-explored source-free supervised transfer learning scenario, where we are given a collection of related and unrelated source models to improve the performance of a new model on a labeled target task. Crucially, we only have access to the target data. *ii)* We propose a framework (shown in Figure 1) that can select several related source models (the number of which is set *a priori*), and utilize their knowledge on new tasks in both *white-box* and *black-box* scenarios. We treat these selected models as 'recycling materials' and reuse them to help learn a better model for the target tasks. *iii)* We perform extensive experiments to highlight the properties of our framework including a specially crafted `CIFAR100`-based experiment and an ablation study.

In the next section, we will discuss related works in relation to the proposed work. In Section 3, we define our problem setting and introduce the parameter-efficient finetune method we adopt as the basic building blocks in our model. In Section 4, we explain in detail each component of our model recycling framework, including the training of source models, the identification of most transferable source models, and the module-mixing model design. We also provide the rationale for the design choices of each component in the ablation study and the Appendix. Experiments are presented in Section 5 including the results for the ablation study. For example, we show that the proposed source model selection module is able to select similar tasks to adapt to with good performance in Figure 5, and compare our method to other model transferability evaluation methods in Table 5. We also analyze the generalization ability of the proposed model and the visualize the features extracted with the learned model. Finally, conclusions are presented in Section 6.

## 2 Related Work

**Multi-source Supervised Transfer Learning**  Tong et al. (2021) presents a transferability measure that is characterized by the sample sizes, model complexity and the similarities between source and target tasks. Then, they use the transferability measure to form a convex combination of the predictions from different models. Most of the works in this class assume source datasets are available during adaptation (Li et al., 2020a; Jin et al., 2021; Xu et al., 2018; Tong et al., 2021; Li, 2022). Alternatively and most similar to our setting, Lee et al. (2019); Wu et al. (2024) do not assume the availability of source data. Specifically, Lee et al. (2019) avoid finetuning the source models by leveraging (maximal) correlation analysis and conditional expectation operators to build a classifier from the combined weighting of the feature functions from the source networks. The inexpensive weighting and lack of finetuning makes it efficient and effective at adapting to multiple source models in the few-shot regime. Our work is different in that we use a convex combination of feature representations *and* task-specific module parameters.

**Source-Free Unsupervised Domain Adaptation**  Li et al. (2020b); Kurmi et al. (2021); Hou & Zheng (2021) focus on generating data for source domains to accommodate already existing unsupervised domain adaptation methods. For instance, Kurmi et al. (2021) generates proxy source samples by treating the trained source classifier as an energy-based model along with a parametric data generative neural network. Chen et al. (2021); Liu & Zhang (2021); Xiong et al. (2021) leverage self-supervised knowledge distillation to finetune source models by adopting a mean-teaching framework (Tarvainen & Valpola, 2017). Liang et al. (2020) use information maximization and self-supervised pseudo-labeling to adapt the target domain to pre-trained source models. Concerning the potential of utilizing diverse knowledge from multiple domains, Liang et al. (2021b); Ahmed et al. (2021b); Dong et al. (2021); Han et al. (2023) explore the possibility of multi-source data-free adaptation. For example, Ahmed et al. (2021b) extends the work from Liang et al. (2020) by combining the source models with trainable aggregated weights. Dong et al. (2021) introduces a confident-anchor-induced pseudo label generator with multiple source models to provide more reliable pseudo labels for target data. Liang et al. (2021b) studies a challenging scenario where only black-box source models are available during adaptation. Most works in this class assume there is no labeled target data, while we discuss the case when target data is labeled (Fang et al., 2024).

**Averaging Model Weights**  Weighted averaging is widely adopted in optimization approaches. Stochastic Weight Averaging aggregates weights along a single optimization trajectory (Izmailov et al., 2018). Matena & Raffel (2021) merge models that are fine-tuned on different text classification tasks with the same pre-trained initialization. Wortsman et al. (2022) focus on averaging weights across independent runs of models on the same dataset. In their "cross-dataset soup", models are trained on different datasets and adapted to target tasks by learning a set of averaging weights. Another similar work Ram'e et al. (2022) also reuses multiple pretrained foundation models to adapt to a new task. The difference is that they finetune the pretrained foundation models on the new task before averaging their weights to create the final model, and assumes that all the pretrained models have the same architecture. Our work is different in that new modules allocated for target tasks are *trained together* with the mixing weights to learn task-specific features. Further, we also discuss how to mix features from different models when their dimensions are different in a black-box scenario.

**Finetuning Models with Task-Specific Modules**  As deep learning models grow in size (He et al., 2015; Dosovitskiy et al., 2020; Radford et al., 2019), storing and finetuning the whole model becomes exceedingly challenging, due to needing expensive computational resources. Therefore, there is a line of work proposing that instead of finetuning all the parameters in a pre-trained model, one can instead partition the network into a frozen backbone model and task-specific modules for finetuning. The idea has been applied to generative models for synthesizing images (Perez et al., 2018; Cong et al., 2020; Verma et al., 2021), discriminative models for image classification (Verma et al., 2021), and finetuning for downstream tasks with large language models (Houlsby et al., 2019; Li & Liang, 2021). This concept is used in building our white-box parameter-efficient models.

**Model Transferability Evaluation** There are studies focused on assessing the transferability of trained models for a specific task. For instance, the pioneer work LEEP (Nguyen et al., 2020), measures the log expectation of the empirical predictor by estimating the joint distribution across pretrained labels and the target labels. It features fast selection speed; however, it requires the availability of the source models' classification heads. LogME proposes to estimate the maximum value of label evidence given features extracted by trained models and obtained the Logarithm of Maximum Evidence (LogME) measure (You et al., 2021). It exceeds LEEP and NCE (Tran et al., 2019) in terms of evaluation accuracy, and only uses extracted features and labels to assess source models. However, it still needs extra training to determine the best suited model, which makes this approach impractical when the model pool is too large. Guo et al. (2023) recognizes a similar challenge as in our work, which is how can a user search for useful models in the *learnware* market. They identify useful models simply by comparing user requirements with model specifications, without running models. However, they need the model developers to submit specifications of their model, which involves generating a feature matrix that describes their data with a public feature extractor. In practice, the model specification details might not always be available. Our choice of model selection features a non-parametric $k$-NN method, which offers a trade-off between selection efficiency and accuracy. We note that the model selection module of our framework can be used interchangeably with the aforementioned methods. Note however that several factors need to be considered when choosing which one to use, including the size of the model pool, the transferability accuracy demand of the users, and also, whether the source model information is limited.

## 3 Background

### 3.1 Problem Setting

In this work, we aim to address the challenge of adapting a collection of classification models trained on different source tasks to a new target task. The goal is to optimize the classification accuracy of the target task. Specifically, consider we have $N$ source models $\{\boldsymbol{h}_{s_1}, \boldsymbol{h}_{s_2}, \cdots, \boldsymbol{h}_{s_N}\}$ corresponding to $N$ source tasks $\{\mathcal{T}_{s_1}, \mathcal{T}_{s_2}, \cdots, \mathcal{T}_{s_N}\}$, and that we also have a target task $\mathcal{T}_t$ with a labeled dataset $\{(\mathcal{X}_t, \mathcal{Y}_t)\}$. Importantly, during training for $\mathcal{T}_t$, we only have access to the target task dataset $\{(\mathcal{X}_t, \mathcal{Y}_t)\}$ and source models $\{\boldsymbol{h}_{s_1}, \boldsymbol{h}_{s_2}, \cdots, \boldsymbol{h}_{s_N}\}$. Each source model consists of a feature extractor $\boldsymbol{f}$ and the classification head $\boldsymbol{c}$. We denote the source model as $\boldsymbol{h}_{s_n} := (\boldsymbol{f}_{s_n} \circ \boldsymbol{c}_{s_n})$, with $\boldsymbol{f}_{s_n} : \mathcal{X}_t \to \mathbb{R}^{d_{s_n}}$ being the feature extractor with output feature vector of dimension $d_{s_n}$, $\boldsymbol{c}_{s_n} : \mathbb{R}^{d_{s_n}} \to \mathbb{R}^{\alpha}$ being a $\alpha$-way classifier and $\circ$ denoting function composition. The goal is to learn a classification model $\boldsymbol{h}_t$ with all the available knowledge. In practice, source and target tasks do not need to be restricted to having the same number of classes since we only use source models as feature extractors.

### 3.2 Efficient Feature Transformation for Task-Specific Modules

In the white-box setting, we propose to build models with task-specific modules for source and target tasks because they are memory efficient and less likely to overfit on small datasets. Specifically, we adopt a task-specific module design termed efficient feature transformation (EFT) (Verma et al., 2021) for our source and target models. EFT proposes appending a small convolutional transformation to each convolutional layer's output feature maps. The transformation involves two kinds of convolutional kernels that capture spatial features within groups of channels (*group-wise filter*) and features across channels at every pixel in the feature map (*point-wise filter*). Given the feature maps $F \in \mathbb{R}^{M \times Q \times K}$ of a layer, with $K$ being the number of feature maps, and $M$ and $Q$ being the spatial dimensions of each feature map, to apply a *groupwise filter* $\omega_i^s \in \mathbb{R}^{3 \times 3 \times a}$, we need to first split $F$ into $K/a$ groups of $a$ feature maps. We then convolve a unique filter $\omega_i^s$ with each group and get a new group of feature maps $H_i^s \in \mathbb{R}^{M \times Q \times a}$ (*i.e.*, each group of feature maps has a unique spatial filter). Subsequently, we concatenate all the $K/a$ new feature maps $H_i^s$ into $H^s \in \mathbb{R}^{M \times Q \times K}$, which has the same dimension as old feature maps $F$. To apply a *point-wise filter* $\omega_i^d \in \mathbb{R}^{1 \times 1 \times b}$, the same procedures as above need to be applied to split $F$ into $K/b$ groups of $b$ feature maps to create $H^d \in \mathbb{R}^{M \times Q \times K}$. The final feature maps are $H = H^s + \gamma H^d$, where $\gamma \in \{0, 1\}$ indicates if the point-wise filters are used or not. By setting $a \ll K$ and $b \ll K$, the amount of trainable parameters is substantially reduced. For instance,

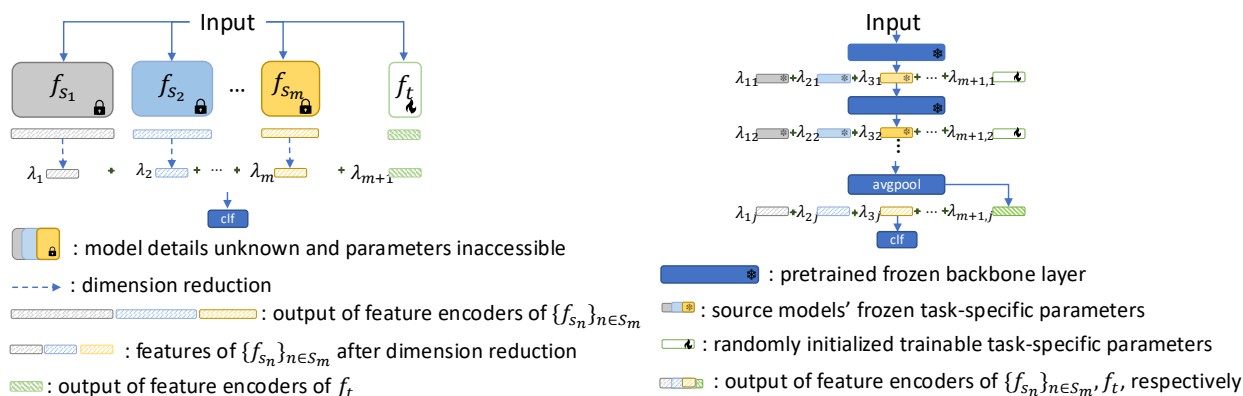

Figure 2: Proposed module-mixing model. Left (black-box): The output features from the source models are first reduced to match the feature dimension of the target model, and then combined with the target features as in equation 4; Right (white-box): For each task-specific layer, modules from source models (frozen during training), and a randomly initialized EFT module are combined as in equation 3. The output features of the source models and that of the new target model are also aggregated as in equation 4.

using a ResNet18 backbone, $a = 8$, and $b = 16$ results in 449k parameters per new task, which is 3.9% the size of the backbone. Additional details can be found in the Appendix A.1 and in Verma et al. (2021).

# 4 Proposed Method

To highlight the flexibility of the framework, we discuss both white-box and black-box assumptions for our source models. We provide a parameter-efficient solution for source model generation. However, we emphasize that the framework is not restricted to using this design, which is why we provide a more flexible setting for the black-box scenario, where we use different pre-trained APIs as our source models. Within both scenarios, the adaptation model in Figure 2 is obtained through two steps. *i)* Given a large pool of $N$ pre-trained source models, it is not practical nor efficient to learn (transfer) from all of them for a new task $\mathcal{T}_t$. Therefore, we first aim to select a subset of $m$ models that are likely trained on related source domains, from which $\mathcal{T}_t$ can learn to improve performance. *ii)* After the selection process, instead of fine-tuning the pre-trained models directly, we propose a method dubbed *module-mixing* for target task adaptation.

**White-box scenario**: all models are trained with a modular architecture sharing a common pre-trained backbone. For each task-specific layer, we perform a convex combination on *a)* all the task-specific modules from selected models and a randomly initialized task-specific module; *b)* the output of the selected source feature extractors and that of the new model.

**Black-box scenario**: we only have access to the output features from source models, and we are transferring to a target model that may have a different (and potentially smaller) architecture from the source models. Specifically, we first reduce the dimensionality of the source model features to match that of the target model, and then aggregate them using a convex combination.

## 4.1 Source Model Generation

For the white box scenario in which we can control the source model generation, we propose using EFT to build a large library of source models to keep computation and memory requirements under control. Assume there are labeled datasets $\{(\mathcal{X}_{s_n}, \mathcal{Y}_{s_n})\}$ for tasks $\mathcal{T}_{s_n}$, $n \in \{1, \cdots, N\}$, with $\mathcal{X}_{s_n}$ and $\mathcal{Y}_{s_n}$ being the feature and label domains, respectively. The feature extractor $\boldsymbol{f}_{s_n}$ consists of a backbone $\phi_{bb}$ that is shared amongst all the source models and a set of task-specific modules $\{\boldsymbol{\theta}_{s_n,j}\}_{j=1}^J$, with $j$ being the task-specific layer index and $J$ being the total number of task-specific layers in the model. Below we write $\boldsymbol{\theta}_{s_n}$ for simplicity. We

train model $\boldsymbol{h}_{s_n}$ by minimizing a standard cross-entropy loss:

$$\boldsymbol{\theta}_{s_n}^*, \boldsymbol{c}_{s_n}^* = \underset{\boldsymbol{\theta}_{s_n}, \boldsymbol{c}_{s_n}}{\arg\min} \ \mathcal{L}(\boldsymbol{\phi}_{bb}, \boldsymbol{\theta}_{s_n}, \boldsymbol{c}_{s_n}; \mathcal{X}_{s_n}, \mathcal{Y}_{s_n})$$

$$= -\mathbb{E}_{(x,y)\in\mathcal{X}_{s_n}\times\mathcal{Y}_{s_n}} \sum_{i=1}^{\alpha} t_i \log(\delta_i(\boldsymbol{h}_{s_n}(x))),$$

where $\boldsymbol{\phi}_{bb}$ is frozen during training, $\boldsymbol{\theta}_{s_n}^*$ and $\boldsymbol{c}_{s_n}^*$ are the optimized task-specific modules and classifier, respectively, $t_i = 1$ for the corresponding class $i$ in one-hot encoding (of $\alpha$ classes), and $\delta_i(\boldsymbol{o}) = \frac{exp(\boldsymbol{o}_i)}{\sum_{j=1}^m exp(\boldsymbol{o}_j)}$ is $i$-th element of the softmax activation output vector $\boldsymbol{o} = \boldsymbol{h}_{s_n}(x)$. We also give further training details, such as the learning rate and batch size in Appendix A.2 Hyperparameter settings.

## 4.2 Selection of Related Source Models

Given the target training dataset $\{(\mathcal{X}_t, \mathcal{Y}_t)\}$, we can extract a dataset of features by collecting the output of the feature extractors $\{\boldsymbol{f}_{s_n}\}_{n=1}^N$. Each feature dataset is denoted as $D_{t,s_n} := \{\boldsymbol{f}_{s_n}(\mathcal{X}_t), \mathcal{Y}_t\}$, for $n \in \{1, \cdots, N\}$. The goal is to select the top-$m$ corresponding source models (denoted as $S_m$) with the best $k$-NN classifier validation accuracy via data from each $D_{t,s_n}$. The rationale is that a higher validation accuracy indicates that the feature extractor used for extracting features was trained on a domain more related to the target task. We consider different values of $m$ in the experiments. Below are the details of our $k$-nearest neighbor ($k$-NN) (Cover & Hart, 1967) classifier. We denote the set of nearest neighbors of $\boldsymbol{f}_{s_n}(\boldsymbol{x})$ as $S_{\boldsymbol{x},s_n}$. In the experiment we set $k = 5$.

For a given validation point $\boldsymbol{x}$, we obtain $\boldsymbol{f}_{s_n}(\boldsymbol{x}_i'') \in S_{\boldsymbol{x},s_n}, i \in \{1, \cdots, k\}$ that satisfies

$$\text{dist}(\boldsymbol{f}_{s_n}(\boldsymbol{x}), \boldsymbol{f}_{s_n}(\boldsymbol{x}')) \geq \max\{\text{dist}(\boldsymbol{f}_{s_n}(\boldsymbol{x}), \boldsymbol{f}_{s_n}(\boldsymbol{x}_i''))\}, \quad \forall (\boldsymbol{f}_{s_n}(\boldsymbol{x}'), \boldsymbol{y}') \in D_{t,s_n} \setminus S_{\boldsymbol{x},s_n}, \tag{1}$$

where $\text{dist}(\cdot, \cdot)$ is a distance metric. We use $D_{t,s_n} \setminus S_{\boldsymbol{x},s_n}$ to represent the subset of $D_{t,s_n}$ excluding $S_{\boldsymbol{x},s_n}$. In this work, we employ the Euclidean distance:

$$\text{dist}(\boldsymbol{f}_{s_n}(\boldsymbol{x}), \boldsymbol{f}_{s_n}(\boldsymbol{x}')) = \left( \sum_{i=1}^d |\boldsymbol{f}_{s_n}(\boldsymbol{x})_i - \boldsymbol{f}_{s_n}(\boldsymbol{x}')_i|^2 \right)^{\frac{1}{2}}.$$

Moreover, the $k$-NN classifier $g_{\text{KNN}}$ is defined as

$$g_{\text{KNN}} = \text{mode}(\boldsymbol{y}'' : (\boldsymbol{f}_{s_n}(\boldsymbol{x}_i''), \boldsymbol{y}'') \in S_{\boldsymbol{x},s_n}), \tag{2}$$

where $\text{mode}(\cdot)$ takes the label value that occurs most frequently in the set $S_{\boldsymbol{x},s_n}$. $g_{\text{KNN}}$ will predict the label for each validation point $\boldsymbol{x}$ based on a majority voting scheme with the nearest neighbors in $S_{\boldsymbol{x},s_n}$.

**Remarks** $k$-NN leverages the majority voting of nearest neighbors when predicting the class labels. A higher $k$-NN score means most of the nearest neighbors have the same label as the sample's ground truth, which indicates the feature space created by the source feature extractor presents the classes in the target task better. Furthermore, when selecting useful sources, our main concern is to find a solution that is *flexible* to both white-box and black-box scenarios, while *maintaining its efficiency* when having to deal with selecting from a large pool of models. $k$-NN, with its non-parametric property, offers insights of how transferable the source models are to the target tasks without any training, which makes it an efficient searching method.

## 4.3 Module-Mixing with Distance Correlation Loss

**White-box Scenario** In this setting, the architecture for all models is defined in advance for convenience. After selecting the top-$m$ source models, we will use them to build the module-mixing model for the target task $t$. As mentioned above, each pre-trained feature extractor $\boldsymbol{f}_{s_n}$ has task-specific weights $\{\boldsymbol{\theta}_{s_n,j}\}_{j=1}^J$. For each task-specific layer $j$, the weights for the target task $t$ are

$$\boldsymbol{\theta}_{t,j} = \lambda_{m+1,j}\boldsymbol{\theta}_{new} + \sum_{n\in S_m} \lambda_{n,j}\boldsymbol{\theta}_{s_n,j}, \tag{3}$$

with $\boldsymbol{\theta}_{new}$ being a randomly initialized new module, $\boldsymbol{\theta}_{s_n,j}$ the frozen task-specific module of layer $j$ of a selected source model from $S_m$, and $\sum_{n=1}^{m+1} \lambda_{n,j} = 1, \forall j \in \{1, \cdots, J\}$. Further, the combined feature representation of the module-mixing model is

$$\boldsymbol{f}(\cdot) = \lambda_{m+1,J+1} \boldsymbol{f}_t(\cdot) + \sum_{n \in S_m} \lambda_{n,J+1} \boldsymbol{f}_{s_n}(\cdot), \tag{4}$$

with $\{\boldsymbol{f}_{s_n}(\cdot)\}_{n \in S_m} \in \mathbb{R}^{\cdot \times d_{s_n}}$ and $\boldsymbol{f}_t(\cdot) \in \mathbb{R}^{\cdot \times d_t}$ being the features (outputs before the classification head) of selected source models and that of the target model, respectively, and $\sum_{n=1}^{m+1} \lambda_{n,J+1} = 1$. Note that $\{\lambda_{n,j}\}_{n=1}^{m+1}, j \in \{1, \cdots, J+1\}$, denoted as $\boldsymbol{\lambda}_t$ for simplicity, are parameters that can be adjusted at initialization and then learned. By setting the initial values of $\boldsymbol{\lambda}_t$, we can control the importance of each task. For instance, we can assume having no prior information on the task importance by initializing $\boldsymbol{\lambda}_t$ in all layers with a uniform distribution as done below in the experiments.

**Black-box Scenario**  We now assume the source models are all black-box APIs, which only produce the features before the classification head. We choose a much smaller architecture for the target model with feature dimensionality being smaller than that extracted from the black-box APIs. The rationale behind this choice is that we want to prevent overfitting provided that the target task usually has a relatively small dataset. Since the extracted feature dimensions from models can be different, we align the dimensions of the APIs' output features to that of the target model via FastICA (Hyvarinen, 1999; Hyvärinen & Oja, 2000), an efficient algorithm for independent component analysis. Then we combine the features as in equation 4. In the experiments, we show that we can still benefit from such a simplified strategy.

**Distance Correlation Loss**  We expect the new modules for the target task can learn knowledge that is not in the mixing modules from the source tasks. Specifically, if we can encourage the learned features from the new modules to be more independent from that of the source models, we could potentially achieve better performance. One way to do so is to train the target module-mixing model with distance correlation loss, which was proposed by Zhen et al. (2022) as a loss function for improving robustness of neural networks against adversarial attacks.

Distance correlation (DC) (Sz'ekely et al., 2007) is a measure of dependence between random vectors. DC between two random variables $X \in \mathbb{R}^p$ and $Y \in \mathbb{R}^q$ satisfies $0 \leq DC \leq 1$, and $DC = 0$ if and only if $X$ and $Y$ are independent. Further, $DC(X, Y)$ is defined for $X$ and $Y$ in arbitrary dimensions. This property allows one to minimize DC between $\{\boldsymbol{f}_{s_n}\}_{n \in S_m}$ and $\boldsymbol{f}_t$ even when their output feature dimensions $d_{s_n} \neq d_t$. Here we follow the notation by Sz'ekely et al. (2007). We use a stochastic estimate of DC by averaging over minibatches $\boldsymbol{x}$ with $n$ samples each. The objective for minimizing the distance correlation is

$$\frac{\langle \boldsymbol{A}(\boldsymbol{f}_{s_n}; \boldsymbol{x}), \boldsymbol{B}(\boldsymbol{f}_t; \boldsymbol{x}) \rangle}{\sqrt{\langle \boldsymbol{A}(\boldsymbol{f}_{s_n}; \boldsymbol{x}), \boldsymbol{A}(\boldsymbol{f}_{s_n}; \boldsymbol{x}) \rangle \langle \boldsymbol{B}(\boldsymbol{f}_t; \boldsymbol{x}), \boldsymbol{B}(\boldsymbol{f}_t; \boldsymbol{x}) \rangle}}, \tag{5}$$

where $\langle \boldsymbol{A}, \boldsymbol{B} \rangle = \sum_{i,j}^n (\boldsymbol{A})_{i,j} (\boldsymbol{B})_{i,j}$, $\boldsymbol{A}(\boldsymbol{f}_{s_n}; \boldsymbol{x}) \in \mathbb{R}^{n \times n}$ (simplified as $\boldsymbol{A}$) and $\boldsymbol{B}(\boldsymbol{f}_t; \boldsymbol{x}) \in \mathbb{R}^{n \times n}$ (simplified as $\boldsymbol{B}$) are distance matrices computed with $X := \boldsymbol{f}_{s_n}(\boldsymbol{x}) \in \mathbb{R}^{n \times d_{s_n}}$ and $Y := \boldsymbol{f}_t(\boldsymbol{x}) \in \mathbb{R}^{n \times d_t}$, respectively, with

$$a_{k,l} = ||X_k - X_l||, \quad \bar{a}_{k,\cdot} = \frac{1}{n} \sum_{l=1}^n a_{k,l}, \quad \bar{a}_{\cdot,l} = \frac{1}{n} \sum_{k=1}^n a_{k,l},$$

$$\bar{a}_{\cdot,\cdot} = \frac{1}{n^2} \sum_{k,l=1}^n a_{k,l}, \quad A_{k,l} = a_{k,l} - \bar{a}_{k,\cdot} - \bar{a}_{\cdot,l} + \bar{a}_{\cdot,\cdot},$$

where $k, l \in \{1, \cdots, n\}$, and $A_{k,l}$ is the $k^{th}$ row and $l^{th}$ column of $\boldsymbol{A}$. Similarly, we can define $b_{k,l} = ||Y_k - Y_l||$, and $B_{k,l} = b_{k,l} - \bar{b}_{k,\cdot} - \bar{b}_{\cdot,l} + \bar{b}_{\cdot,\cdot}$. We optimize all the module-mixing models with the cross-entropy loss and DC loss as below, with $\sigma$ being a constant trade-off parameter:

$$\mathcal{L}_{total} = \mathcal{L}_{CE} + \sigma \sum_{n \in S_m} \mathcal{L}_{DC}(\boldsymbol{f}_{s_n}(\cdot), \boldsymbol{f}_t(\cdot)), \tag{6}$$

---

**Algorithm 1** Module-mixing Model.

---

**Data:** Training data $D_t = \{\mathcal{X}_t, \mathcal{Y}_t\}$; Selected source models $\{\boldsymbol{f}_{s_n}\}_{n \in S_m}$;
**Result:** White-box: Module-mixing model $\boldsymbol{f}$ constructed with equation 4 and equation 3; Black-box: Module-mixing model $\boldsymbol{f}$ constructed with only equation 4

**for** *epoch* $\leftarrow 1$ **to** $C$ **do**
    **for** *each minibatch* $\boldsymbol{x}$ **do**
        **for** $n \leftarrow 1$ **to** $S_m$ **do**
            Calculate distance matrices $\boldsymbol{A}(\boldsymbol{f}_{s_n}; \boldsymbol{x})$ and $\boldsymbol{B}(\boldsymbol{f}_t; \boldsymbol{x})$ in equation equation 5
            Calculate $\mathcal{L}_{DC}(\boldsymbol{f}_{s_n}(\boldsymbol{x}), \boldsymbol{f}_t(\boldsymbol{x}))$ as in equation 5
        **end**
        Calculate $\mathcal{L}_{total}$ as in equation 6
        Update $\boldsymbol{\theta}_{new}, \boldsymbol{c}_t, \boldsymbol{\lambda}_t$ using $\boldsymbol{\theta}_{new}^*, \boldsymbol{c}_t^*, \boldsymbol{\lambda}_t^* = \arg\min \mathcal{L}_{total}(\boldsymbol{h}_t; \mathcal{X}_t, \mathcal{Y}_t)$
    **end**
**end**

---

The objective $\boldsymbol{\theta}_{new}^*, \boldsymbol{c}_t^*, \boldsymbol{\lambda}_t^* = \arg\min \mathcal{L}_{total}(\boldsymbol{h}_t; \mathcal{X}_t, \mathcal{Y}_t)$ indicates that the trainable parameters in $\boldsymbol{h}_t$ are the new modules $\boldsymbol{\theta}_{new}$, the classification head $\boldsymbol{c}_t$ and convex combination parameters $\boldsymbol{\lambda}_t$. The detailed algorithm is presented in Algorithm 1. Overall, our motivation for using distance correlation loss is that it encourages features extracted from previous models to be independent from the features learned from the new tasks; Furthermore, since the output feature dimensions for new and old tasks might not be the same, we need the loss to be able to handle features of different dimensions.

The reasons for using this module-mixing technique are threefold. First, it addresses feature saturation and the issue of plasticity loss that occurs when model weights are fine-tuned over time, which is discussed in Dohare et al. (2021); Ash & Adams (2019). They argue that, in a continual learning setting, if a model is finetuned sequentially on several tasks, the model will likely not benefit from random initialization for later tasks as it will lose plasticity over time. Therefore, our goal is to devise a way to make use of pre-trained models rather than attempting to modify them. Our model ensures that we can learn from previous knowledge while also adding new capacity at the start of training for each new task. Secondly, the flexible nature of the model enables us to investigate whether there is a benefit to learning from more tasks, how to balance training and inference efficiency, and the number of tasks to be reused. The reason is that when the number of selected tasks increases, the amount of trainable parameters of the model grows slowly since only $\boldsymbol{\lambda}_t$ is growing, which is negligible relative to $\boldsymbol{\theta}_{new}$ and $\boldsymbol{c}_t$. Importantly, the last and third reason is that this approach allows us to transfer whether the source models are treated as white- or black-boxes.

## 5 Experiments

**Datasets** We create three main tasks with `CIFAR100` (Krizhevsky, 2009), `Office-31` (Saenko et al., 2010), and the $S^{long}$ stream in `CTrL` (Continual Transfer Learning benchmark) (Veniat et al., 2020). Each dataset represents a unique scenario. With a special task creation scheme for `CIFAR100` (detailed below), we study the cases where the classes in each task overlap or are completely different. With `Office-31`, each task has the same labels but different data distributions. With the challenging `CTrL`, we have a large pool of models from which to start, and source and target tasks come from very distinct datasets and have diverse sample sizes. Moreover, an experiment on `Tiny-ImageNet` (Le & Yang, 2015) is shown in the ablation study.

**Network Architecture and Implementation Details** For $k$-NN source model selection, we set $k = 5$ for all experiments, then select the source models with the top-$m$ highest $k$-NN scores based on the results of each task's validation set. One experiment discussing other settings for $k$ is provided in the Appendix A.5. *(i)* In the white-box setting, for source model generation and target task adaptation, we choose a pre-trained ResNet-18 on ImageNet as our frozen backbone and train the EFT modules with the Adam optimizer. We set $a = 8$ and $b = 16$ for EFT for experiments on `CIFAR100` and `Office-31`, and $a = 2$ and $b = 1$ for EFT on `CtrL`. *(ii)* In the black-box setting, a simple LeNet-5 (LeCun et al., 1998) is used as target model. We provide results for when the source models are (*a*) ResNet-18s with EFT and (*b*) a

pre-trained MAE (He et al., 2021) (written as API in the tables). The DC loss trade-off is set to $\sigma = 0.05$ in all experiments. Implementation details can be found in the Appendix A.2, and the source code can be found at `https://anonymous.4open.science/r/module_mixing-00C7`.

**Baselines** (*a*) **Independent Model:** Add randomly initialized EFT modules to a pre-trained frozen backbone and train the model solely with target task data. (*b*) **Multi-Source SVM**: Extract features from $m$ randomly selected source models with the target training set, then concatenate the $m$ set of extracted features and use them to train an SVM model (Cortes & Vapnik, 1995). (*c*) **Maximum Correlation Weighting (MCW)**: Learn the maximum correlation parameters of $m$ randomly selected source feature extractors with one pass of the target training data (Lee et al., 2019). (*d*) **Finetune Source**: Take the source model selected by $k$-NN with $m = 1$, and finetune the corresponding task-specific modules. (*e*) **Cross-dataset Soup**: Train an independent model for the target task first, and then train the mixing weights with selected source models (Wortsman et al., 2022) with top-$m$ $k$-NN scores. (*f*) **Model Stacking**: Make predictions with the selected top-$m$ source models and an independent model for the target task, then use those predictions as input features in a logistic regression model. (*g*) **DECISION**: Freezes the last classification layers of source models and jointly optimize the source feature encoders together with their aggregated weights. The source classification layers are used to generate pseudo labels for target data (Ahmed et al., 2021a). (*h*) **DATE (SHOT)**: Utilizes a source-similarity transferability module and a proxy discriminability perception module for weight determination for source models (Han et al., 2023). (*i*) **DINE**: First distills knowledge from the source predictor to a customized target model, then fine-tune the distilled model to further fit to the target domain (Liang et al., 2021a).

**MCW**, **DECISION**, **DATE**, **DINE** are all previous state-of-the-art (SOTA) methods. However, only **MCW** is designed for the same multi-source data-free supervised transfer learning setting (target task has labels). **DECISION** and **DATE** have the most similar problem setting as ours, which is multi-source data-free unsupervised domain adaptation (target task does not have labels). Furthermore, **DINE** is proposed for adapting to single-source and multi-source black-box models when target task does not have labels. The results of **DECISION**, **DATE** and **DINE** are obtained by strictly conforming to their public source code.

Though **DECISION**, **DATE (SHOT)**, **DINE** achieve great performance under each of their settings, they all assume close-set label spaces between source and target domains. Besides, due to their unsupervised nature, they all use pseudo labels predicted by the source models to train the target models. According to Yi et al. (2023), when the noise rate in pseudo labels is high at the beginning of the training, the target model will quickly remember the noise due to confirmation bias. All the above reasons explain the performance deterioration of these methods under our setting.

## 5.1 White-box Scenario

**CIFAR100** We create a set of 40 tasks with `CIFAR100`. Different tasks in the collection may share some of the same classes. However, we ensure that the overlapping classes do not have the same data samples by splitting data for each class into two even parts and only allowing each class to exist in tasks twice; this allows us to create a scenario where each task is private and has unique data samples. Details of each task are shown in the Appendix A.2.1. We run each experiment 3 times. For each run, we randomize the list of 40 tasks and pick the first ten of the list as the starting source task pool. For the rest of the tasks, $11 - 40$, we proceed as follows. First, for target task 11, we randomly select a subset of size $e = \{40, 80, 160, 320\}$. With each subset, we select $m = \{1, 3, 5\}$ related source tasks from the pool to train our new module-mixing model. Before moving to target task 12, we train an independent model (via EFT) for task 11 with all available data. This is then repeated for all the other tasks, $12 - 40$. Performance is reported for different values of $m$ and $e$ averaged over tasks $11 - 40$. This is done to obtain results for target tasks of different sizes while making sure all source models are trained with sufficient data.

**Influence of the Number of Source Models** It is important to note that each task has a different optimal selection of $m$ since the number of related tasks with positive transfer is different for each task. However, for the sake of simplification, we report the results with a fixed number of source models (with $m = 1, 3, 5$) for each task. In Figure 3 (Left), we observe that with all of the chosen $m$, we obtain better

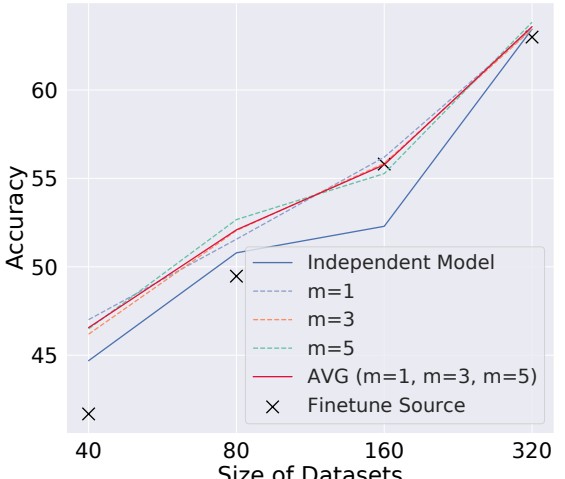 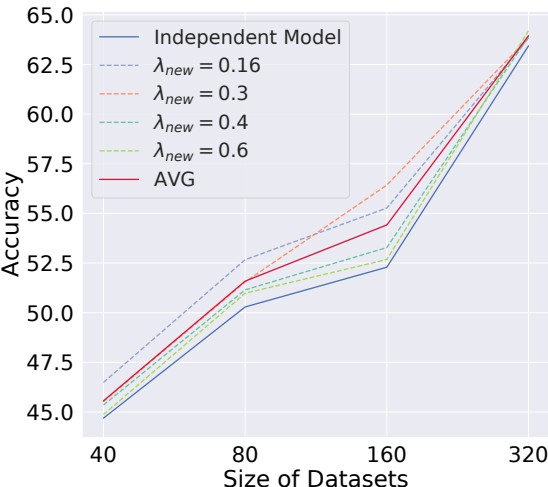

Figure 3: Left: Analysis of the influence of dataset size with respect to the number of source models adapted on `CIFAR100`. Right: Analysis of the influence of dataset size $e$ with respect to different mixing weight initialization on `CIFAR100` ($m = 5$ for all the experiments). $\lambda_{new}$ on the right panel represents the initial value of the new random initialized modules.

results than directly finetuning a source model and training from scratch. Moreover, a wider gap is observed between the independent model and the average performance of our module-mixing model with varying numbers of source models as the size of the datasets decreases, indicating that one potentially gets better performance gains from our method compared to training an independent model. Further, directly finetuning a source model may result in overfitting, especially when the dataset size is small. Alternatively, our model effectively reduces the chances of overfitting.

**Influence of Different Mixing Weight Initialization**   We also show the effect of the mixing weights' initialization on overall performance. We consider all the source tasks equally, hence no extra information on which one is more important from the start. Hence, we initialize their mixing weights with equal values and only change the weight values of the new random initialized modules, *i.e.*, $\lambda_{new} = \lambda_{m+1,J+1}$. For simplicity, all layers share the same set of weights at initialization. In Figure 3 (Right), for $m = 5$, we observe that the average accuracy for different mixing weight initialization is always better than the independent model, albeit with particular initialization values one may get better performance than the average.

**Office-31**   `Office-31` consists of 31 categories of images originating from 3 domains: Amazon, DSLR and Webcam, including common objects in everyday office settings. Webcam consists of 795 low-resolution images with noise; the Amazon domain has 2817 images from online merchants; DSLR images are captured with a digital camera, and all 423 images are of high resolution and low noise. With this dataset, we show how our model behaves to covariate shift, when all the tasks have sufficient data and share the same labels.

In this context, Finetune Source becomes a more competitive baseline now that it does not suffer from serious overfitting issues. From Table 1, our model $m = 1$ guarantees that it has better if not equal performance compared to finetuning the same source model. Moreover, we achieve a $7.75\%$ performance increase compared to Finetune Source when adapting to both two source models training with DC loss. We also see that on average, learning from more source models that are trained on a related domain yields better positive transfer, specifically, we observe a $2.78\%$ increase from going from $m = 1$ to $m = 2$ when training with DC loss. Since MCW is designed as a fast-to-adapt and lightweight model that does not introduce too many trainable parameters, it loses its advantage in this setting when compared to other finetuning models, and even multi-source SVM when the source models are trained on closely related domains. As for cross-dataset soup and ensemble methods like model stacking, they work better when all the models are trained on the same datasets (or different splits of the same dataset).

Table 1: Accuracy comparison on `Office-31` in the white-box scenario. The highest accuracy is marked in bold. We also compare the results training with and without distance correlation loss for our model.

| Method | $A, D \to W$ | $A, W \to D$ | $W, D \to A$ | AVG |
|---|---|---|---|---|
| Model stacking | 58.75 | 68.62 | 64.31 | 63.89 |
| Cross-dataset soup | 11.88 | 21.78 | 10.84 | 14.83 |
| Multi-source SVM | 90.11 | 81.82 | 84.98 | 85.64 |
| MCW | 64.84 | 62.12 | 76.79 | 67.92 |
| DATE (SHOT) | 60.74 | 52.74 | 20.43 | 44.63 |
| DECISION | 59.22 | 50.42 | 21.35 | 43.66 |
| Independent | 89.24 | 75.62 | 71.73 | 72.34 |
| Finetune Source | 91.25 | 88.24 | 82.69 | 87.39 |
| $m = 1$ (w/o $\mathcal{L}_{DC}$) | **97.50** | 88.24 | 91.52 | 92.42 |
| $m = 2$ (w/o $\mathcal{L}_{DC}$) | 95.00 | **94.12** | 91.17 | 93.43 |
| $m = 1$ (w/ $\mathcal{L}_{DC}$) | 95.00 | 90.20 | 91.87 | 92.36 |
| $m = 2$ (w/ $\mathcal{L}_{DC}$) | **97.50** | **94.12** | **95.05** | **95.14** |

Table 2: Accuracy comparison on `CTrL` in the white-box scenario. For our model, besides showing the effect of the distance correlation loss, we also show the importance of a hyperparameter grid search when the model pool is filled with models that are trained on very different source datasets.

| Method | $m = 1$ | $m = 3$ | $m = 5$ | AVG |
|---|---|---|---|---|
| Model stacking | 42.53 | 34.02 | 34.31 | 36.95 |
| Cross-dataset soup | 47.73 | 43.83 | 39.43 | 43.66 |
| Multi-source SVM | 44.34 | 28.21 | 26.59 | 33.08 |
| MCW | 42.73 | 45.02 | 54.09 | 47.28 |
| Independent | - | - | - | 45.77 |
| Finetune Source | - | - | - | 54.26 |
| DATE (SHOT) (knn) | 42.45 | 43.72 | 42.84 | 43.00 |
| DECISION (knn) | 42.56 | 45.24 | 45.68 | 44.49 |
| Multi-source SVM (knn) | **68.48** | 56.96 | 48.33 | 57.92 |
| MCW (knn) | 66.59 | **72.89** | **72.91** | **70.80** |
| Ours (w/o $\mathcal{L}_{DC}$) | 56.40 | 55.13 | 52.09 | 54.54 |
| Ours (w/$\mathcal{L}_{DC}$) | 56.48 | 55.35 | 52.79 | 54.87 |
| Ours (w/$\mathcal{L}_{DC}$ grid search) | 67.26 | 70.24 | 68.71 | 68.73 |

**CTrL** The $S^{long}$ stream from `CTrL` is a collection of 100 tasks created from five well-known computer vision datasets: `CIFAR10`, `CIFAR100`, `SVHN` (Netzer et al., 2011), `MNIST` (LeCun et al., 2010), and `Fashion-MNIST` (short as `FMNIST`) (Xiao et al., 2017). Each task in $S^{long}$ is constructed by first randomly selecting a dataset, then five classes of the chosen dataset, and finally, a large task (containing 5000 training samples) or a small task (containing 25 training samples). During tasks $1 - 33$, the fraction of small tasks is 50%; this increases to 75% for tasks $34 - 66$, and to 100% for tasks $67 - 100$. More details of the dataset can be found in the Appendix A.2.2. In our experiments, we use the first 60 tasks as our starting pool of source models and gradually add newly trained models into the collection. We report the average test accuracy on the last 40 tasks. This is a challenging problem not only because it simulates a realistic setting that starts with a large collection of pre-trained tasks for repurposing, but also because the last 40 tasks have only 25 training samples each, which could cause serious overfitting problems. We perform a simple grid search on hyperparameters (learning rate and weight decay) and compare them to having the same settings for all tasks. Grid search and other experimental details are in the Appendix A.2. Results show that grid search is necessary when tasks come from very different datasets. In Table 2, we also show results of an extension of MCW and multi-source SVM with our $k$-NN source model sampler. The results show the effectiveness

Table 3: Accuracy comparison on `CIFAR100` in the black-box scenario. For our model, we also compare the results training with and without distance correlation loss.

| Method | $m = 1$ | $m = 3$ | $m = 5$ | API | AVG |
|---|---|---|---|---|---|
| Model stacking | 71.32 | 70.97 | 70.36 | - | 70.88 |
| Cross-dataset soup | 39.71 | 37.17 | 35.93 | - | 37.60 |
| Multi-src SVM | 54.76 | 51.48 | 47.60 | - | 51.28 |
| MCW | 50.46 | 52.75 | 59.34 | - | 54.18 |
| DINE | 40.41 | 41.23 | 39.76 | - | 40.47 |
| Independent | - | - | - | - | 70.39 |
| Ours (w/o $\mathcal{L}_{DC}$) | 71.43 | 71.74 | 71.17 | 69.97 | 71.08 |
| Ours (w/ $\mathcal{L}_{DC}$) | **72.03** | **71.82** | **71.46** | **71.80** | **71.78** |

of our $k$-NN source model sampler, helping to achieve a 23.52% increase for MCW compared to learning from randomly sampled source models. However, we still suffer from the overfitting problem mainly on tasks created with `SVHN`, which explains the gap between our method and MCW with $k$-NN sampler.

## 5.2 Black-box Scenario

Results on `CIFAR100` (with all data in each task) are shown in Table 3. We also used $k$-NN to select source models for baselines. The results show that we can still benefit from such a simple approach. Additional experiment results on other datasets in this setting can be found in the Appendix A.6.6.

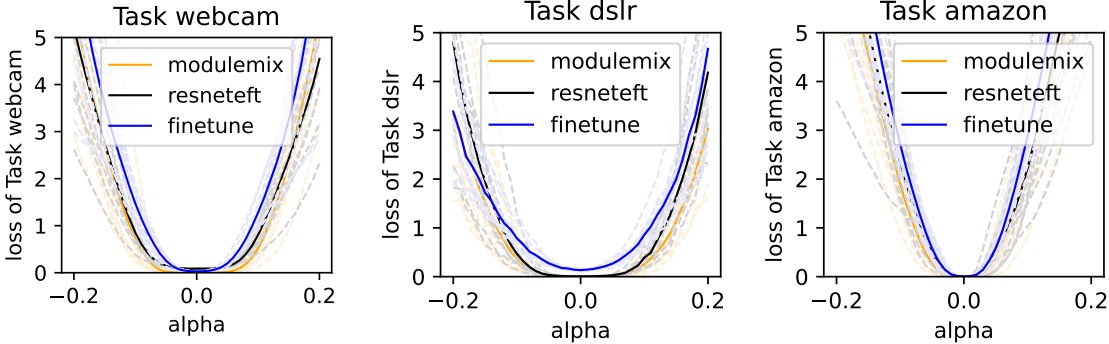

Figure 4: Comparison of weight loss landscapes. The dotted lines shows the loss created with the random directions given the trained model weights.

## 5.3 Insights Behind Module-Mixing Framework Design

Deng et al. (2021); Liebenwein et al. (2021); Cha et al. (2021); Wortsman et al. (2022) have shown that a model with a flatter loss landscape contributes to the generalization of models in domain adaptation and continual learning, thus is more robust to overfitting compared to directly finetune the source model. A similar idea as ours proposed in Wortsman et al. (2022) showed that by averaging weights of multiple finetuned models with the same initialization, the model's optimum falls in a flatter loss/error landscape with overall lower loss/error. Based on Li et al. (2017), the sharpness of loss landscapes correlates well with generalization error.

To show that the module-mixing framework has a better generalization ability and more robust to overfitting than directly finetuning source models, in Figure 4 we show the weight loss landscape of the independent model (resneteft), finetune, and our module-mixing model ($m = 1$) trained on all data from three domains in `Office31`. We follow the procedure from Deng et al. (2021) and create ten random directions given the

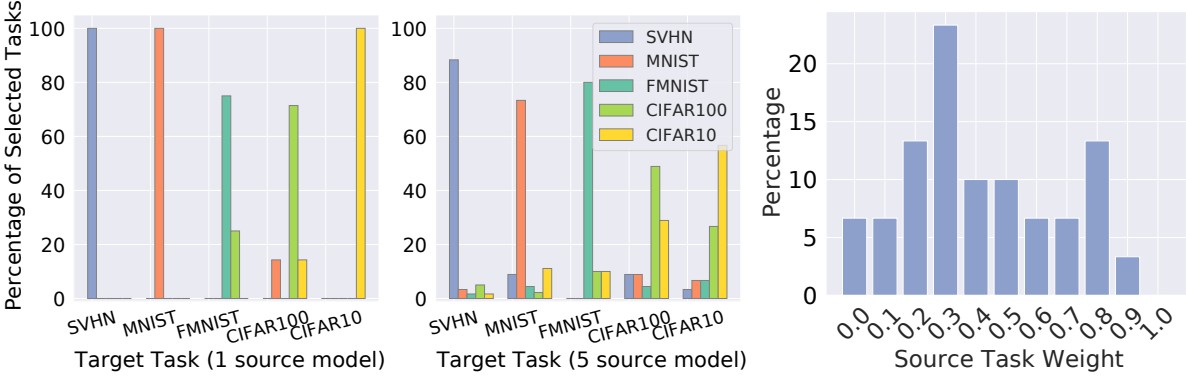

Figure 5: Selection of $m = 1$ (Left) and $m = 5$ (Middle) related tasks via $k$-NN. The $x$-axis shows the membership of the target tasks, while the $y$-axis shows the proportion of selected source tasks for each membership. Right: Module-mixing results for $m = 1$. The $x$-axis is the mixing weight value set for the source model and the $y$-axis is the percentage of tasks in `CIFAR100` for which the highest accuracy is obtained at a given source weight setting.

trained model weights. The solid line is the average over the results from the ten random directions. We can see from the plots that module-mixing model has a flatter loss landscape than the other methods, which explains why the model alleviates the effects of potential overfitting more than directly finetuning the model.

We also provide another plot for module-mixing under black-box setting with limited target data in the Appendix A.6.1. In the figure, we compare the module-mixing model with the Finetune Source model trained on webcam with only 20% of its data. Same conclusions can be drawn from this setting.

## 5.4 Ablation Study

We construct a module-mixing model with the top-1 source model selected via $k$-NN and a target model using only equation 4. In this setting, the mixing weight $\lambda_1$ for the source model features is set as a value in $\{0, 0.1, \ldots, 0.9, 1\}$, and only the target model's modules and the classification head are trained. Thus, an independent model is trained when $\lambda_1 = 0$, whereas only the classification head of a pre-trained source model is trained when $\lambda_1 = 1$. We test on our 30 `CIFAR100` tasks. For each task, one setting for $\lambda_1$ will achieve the highest test accuracy. Each bar in Figure 5 (Right) shows the percentage of tasks that reached peak accuracy at the given task weight setting. We see that the optimum results are mostly obtained (93%) when the mixing weight is non-zero for either source and target models, thus justifying our model design.

Table 4: Ablation Study on `CTrL` dataset when $m = 1$.

| Method | Accuracy |
|---|---|
| Independent model | 45.77 |
| Random (only equation 4) | 45.88 |
| $k$-nearest neighbors (only 4) | 55.68 |
| Random (both equation 4 and equation 3) | 46.61 |
| $k$-nearest neighbors (both equation 4 and equation 3) | **56.40** |

In Table 4, we examine the advantages of our source model selection and layer-wise module-mixing design on `CTrL` (with $m = 1$ for adaptation). Alternatively, we consider selecting source models at random (instead of $k$-NN) and only using the convex combination of features in equation 4 (instead of both features and module parameters). We see that with random selection and only feature aggregation, we get an insignificant gain of 0.11% relative to the independent model, while with $k$-NN selection and only feature aggregation, the gain is higher at 1%. Importantly, the complete approach using equation 4 and equation 3 with k-nearest neighbors model ewselection yields a substantial 9.79% gain relative to that with random source model selection.

Table 5: `TinyImageNet` in the white-box scenario. Selected task names are colored for clarity.

| | | Task1 | Task2 | Task3 | Task4 | Task5 | Task6 | AVG |
|---|---|---|---|---|---|---|---|---|
| | Independent model | 34.55 | 32.64 | 35.07 | 32.47 | 33.16 | 33.25 | 33.52 |
| | Finetune source | 34.38 | 31.77 | 33.51 | 32.73 | 33.59 | 32.90 | 33.15 |
| $m = 2$ | Model stacking | 31.03 | 31.86 | 35.33 | 31.43 | 30.64 | 28.91 | 31.53 |
| | Cross-dataset soup | 28.65 | 32.20 | **37.24** | **34.20** | 33.51 | 32.73 | 33.09 |
| | Multi-source SVM | 28.64 | 25.44 | 25.76 | 24.72 | 27.04 | 23.76 | 25.89 |
| | MCW | 21.76 | 22.32 | 22.40 | 21.68 | 22.16 | 21.68 | 22.00 |
| | DATE (SHOT) | 21.24 | 22.76 | 21.49 | 22.36 | 21.66 | 21.90 | 21.90 |
| | DECISION | 21.34 | 21.93 | 21.36 | 21.58 | 22.46 | 20.96 | 21.60 |
| $m = 1$ | LogME | 35.16 | **34.80** | 35.36 | 32.90 | 35.07 | 33.28 | 34.43 |
| | Selected task | Task2 | Task1 | Task2 | Task5 | Task4 | Task5 | |
| | LEEP | 35.16 | 33.59 | 35.36 | 32.90 | 35.07 | 34.08 | 34.36 |
| | Selected task | Task2 | Task3 | Task2 | Task5 | Task4 | Task3 | |
| | Ours | 35.16 | 33.59 | **36.26** | 32.90 | **35.07** | **35.40** | **34.73** |
| | Selected task | Task2 | Task3 | Task4 | Task5 | Task4 | Task4 | |
| $m = 2$ | LogME | 35.00 | 34.08 | 33.68 | 33.52 | 34.96 | 33.36 | 34.10 |
| | Selected tasks | Task2 / Task6 | Task1 / Task3 | Task2 / Task4 | Task3 / Task5 | Task4 / Task6 | Task4 / Task5 | |
| | LEEP | 35.00 | 34.08 | 33.68 | 33.52 | 34.96 | 34.96 | 34.37 |
| | Selected tasks | Task2 / Task6 | Task1 / Task3 | Task2 / Task4 | Task3 / Task5 | Task4 / Task6 | Task3 / Task5 | |
| | Ours | **36.40** | 34.08 | 33.68 | 33.52 | 34.96 | 33.84 | 34.41 |
| | Selected tasks | Task2 / Task3 | Task1 / Task3 | Task2 / Task4 | Task3 / Task5 | Task4 / Task6 | Task1 / Task4 | |

To show that our method also applies to more complex tasks, we provide an experiment on `Tiny-ImageNet`. We use a randomly initialized ResNet-18 as the backbone. The purpose of this setting is to show that the improvement in performance of our model does not rely on a powerful and closely related backbone model since `Tiny-ImageNet` is a subset of `ImageNet`. We create 6 tasks with 200 classes in the dataset, with 50 classes for each task. Task 1 through 5 have overlapping classes, while Task 6 has no overlapping with other tasks. Task details are presented in Table 17 and 18, where tasks with overlapping classes have the same color-coding. Since the tasks in this setting all have sufficient training data, the newly initialized task-specific modules for the target task are given higher initial module-mixing weights (0.9) to encourage the learning of target task-specific knowledge. For each task, we search for top-$m$ most similar tasks in the other 5 tasks. In Table 5, we also provide the source model selection results, shown in "selected task". The task names have the same colors as in the header of the table to provide more clarity. We observe that tasks with overlapping classes are always picked for each task. For instance, Task 2 overlaps with Task 1 and 3, so when we build our module-mixing model with the top-2 source models to reuse, these tasks are selected.

Furthermore, we compare with LEEP (Nguyen et al., 2020) and LogME (You et al., 2021) in identifying the most transferable source models. Different model transferability evaluation methods are first used to select the top-$m$ transferable models, and then the selected source models are used to build our module-mixing models to transfer to each target task. As is shown in Table 5, our selection method's performance is on par with the baselines, while having a slight edge over them. We also provided the average results and standard deviation over three runs comparing our method to LogME and LEEP in the Appendix A.6.2.

Moreover, we show how well the $k$-NN algorithm selects source models with `CTrL`. In Figure 5, we show selection results with $m = 1$ (Left) and $m = 5$ (Middle). Each bar represents the proportion of selected tasks from each dataset for the 40 tasks. When $m = 1$, target tasks created with `SVHN`, `MNIST`, and `CIFAR10` always pick source models trained from the same datasets. Also, we only have 4 `FMNIST` tasks in the experiments, which explains why the selector picks a high 25% of `CIFAR10` source models. As $m$ increases to 5, tasks

with classes from `SVHN`, `MNIST`, `FMNIST` still almost always pick models created from the same dataset. For `CIFAR10` and `CIFAR100`, since they both contain images of natural objects, they are more similar.

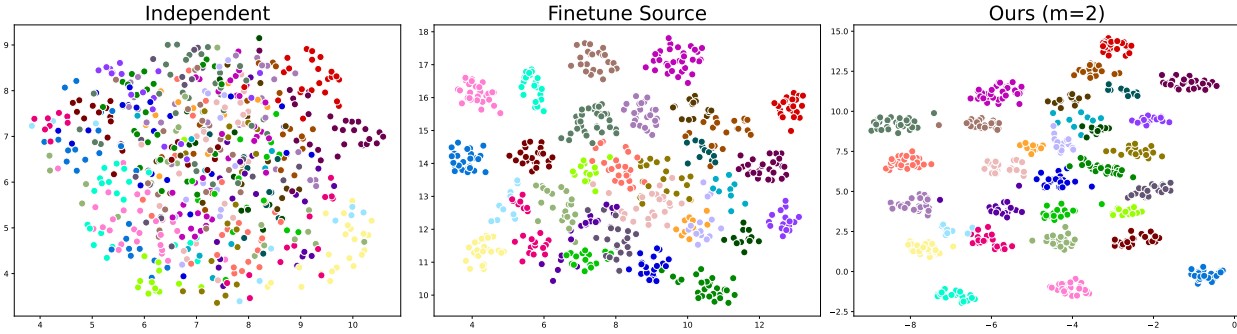

Figure 6: Umap plots on the Webcam domain: each color represents the same label across all three panels.

### 5.4.1 UMAP Plots of Office-31 Domains

Finally, we validate if our model learned meaningful data representations by visualizing the features of training data extracted from the feature extractor of models Independent, Finetune Source and our Module-mixing model, respectively, using UMAP (McInnes et al., 2018) as shown in Figure 6. Learned features with our model are relatively more clustered and separated compared to that for the other two methods. We also provide quantitative results examining the separation of the features in Appendix A.6.4. For the DSLR domain, the same conclusions can be drawn, as shown in Figure 12 in the Appendix A.6.3.

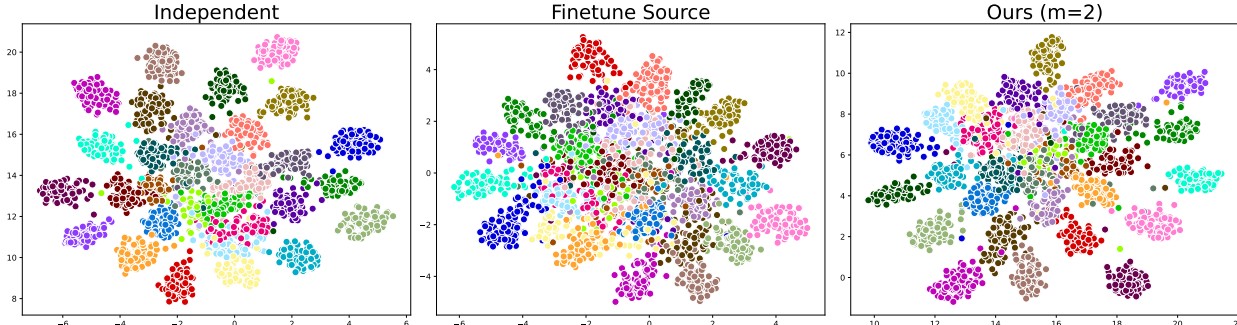

Figure 7: Umap plots on the Amazon Domain with training data. Each unique color represents the same label across all three panels.

For the Amazon domain, in Figure 7, we create scatter plots of the learned UMAP embeddings of the training data, and color samples (points) according to their ground truth labels. All three plots in Figure 7 have good clustering patterns. However, if we color-code the training data points by their respective predicted probabilities of outcomes as in Figure 8, we notice that models Independent and Finetune Source are overconfident about their predictions, while our method also has the effect of preventing the model from predicting the labels too confidently during training. In Figure 9, we visualize the extracted features of the test data after UMAP embedding. We see that the Independent model is less confident about the predictions. While Finetune Source has higher predicted probabilities on the test set, our model has relatively better clustering and is more certain about the predictions compared to Finetune Source. Besides the loss weight landscape analysis, this observation can be seen as another indication that our model has better generalization.

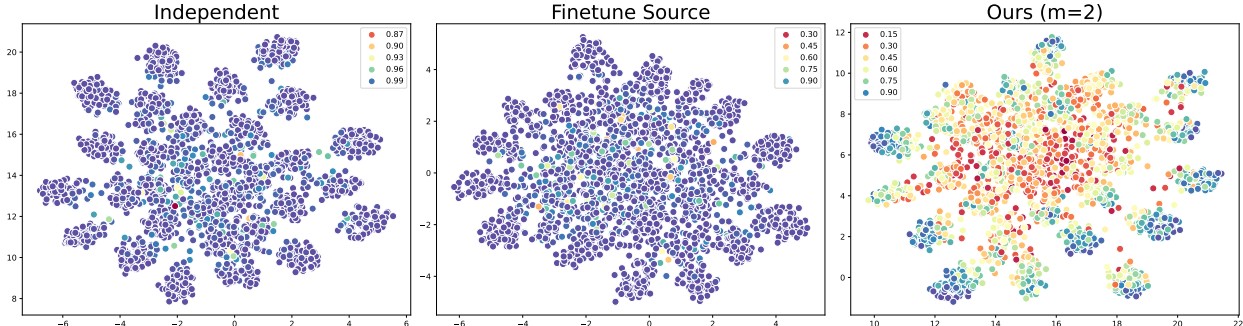

Figure 8: Umap plots on the Amazon domain with training data. The colors represent the predicted probability. Independent and Finetune Source have high overall probability for their predictions, while our method does not. This indicates that our method has better generalization.

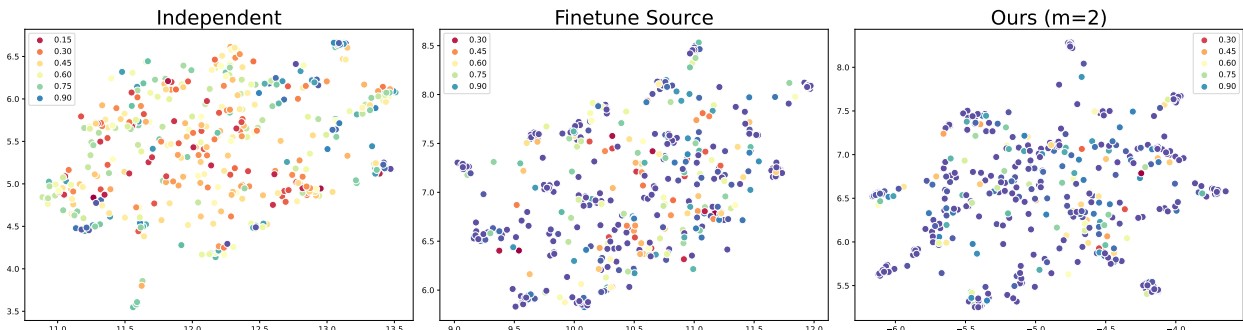

Figure 9: Umap plots on the Amazon domain with testing data. The colors represent the predicted probability. Independent and Finetune Source have lower overall probability for their predictions compared to our method's predictions.

## 6 Conclusion

In this work, we studied an under-explored source-free supervised transfer learning scenario. The proposed framework can efficiently search for related models to adapt to and also can be applied to white-box and black-box source models. To test our module-mixing model and encourage more work in this field, we not only experimented on a classic domain adaptation dataset `Office-31`, but also crafted new datasets based on existing benchmarks, including `CIFAR100`, `CTrL` and `Tiny-ImageNet`. We showed promising improvements compared to the baselines and examined the properties of each part of our framework.

Nevertheless, we acknowledge some limitations of the proposed method. So far, we initialize the model's mixing weights with equal values or do a grid search for the best mixing weights. However, it will be interesting to study how to initialize the mixing weights according to different source models' transferability to the target task. Moreover, we require all models to have task-specific modules of same sizes for the white-box scenario. Therefore, another interesting future work is to see how to build layer-wise module-mixing modules when the task-specific modules are not of the same sizes.

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
