# OpenReview forum: "Model Recycling Framework for Multi-Source Data-Free Supervised Transfer Learning"
_TMLR — Rejected by TMLR_

### Review · Reviewer_uPEN · 2024-06-27

**Summary Of Contributions:**

In this work, the authors tackle the problem of source-free transfer learning. They propose a method for selecting a subset of models from a pool of pre-trained models on various tasks, based on their similarity to the target task. The final predictions are then made by combining the outputs of the selected models.

**Audience:**

Yes

**Broader Impact Concerns:**

I don't have any concerns.

**Claims And Evidence:**

No

**Requested Changes:**

**Fact-checking and literature research (absolutely necessary)**
- “As most existing transfer learning methods require the availability of source data, and thus cannot be directly adapted to the source-free scenario.” This statement is overly general and incorrect. Numerous subfields within transfer learning explore methods that do not require access to source data, ranging from simple fine-tuning ([Oquab 2014, CVPR](https://openaccess.thecvf.com/content_cvpr_2014/papers/Oquab_Learning_and_Transferring_2014_CVPR_paper.pdf)) to source-free test-time adaptation ([Awesome Source-Free Test-Time Adaptation](https://github.com/YuejiangLIU/awesome-source-free-test-time-adaptation)). Moreover, the concept of recycling strategies of several pretrained networks is not that rare as claimed, as evidenced by existing literature ([Rame, ICLR 2023](https://proceedings.mlr.press/v202/rame23a/rame23a.pdf)).
- “We not only experiment on classic domain adaptation datasets.” Contrary to this claim, the authors do not use classic domain adaptation datasets. For a list of commonly used datasets, refer to [DomainBed](https://arxiv.org/abs/2007.01434), and consider testing their method on datasets like PACS or Office-Home.
- The motivation behind the distance correlation loss is unclear, especially since they cite continual learning scenarios, which are not directly applicable here.
- “As shown in Figure 6, learned features with our model are relatively more clustered and separated compared to those for the other two methods.” This conclusion should not be drawn from a UMAP plot, particularly when the distinction is not evident in Figures 6 and 7.
- “Figure 8: UMAP plots on the Amazon domain with training data. The colors represent the predicted probability. Independent and Finetune Source have high overall probability for their predictions, while our method does not.” The conclusion about better generalization based solely on lower prediction confidence is unfounded and lacks adequate justification.

**Proper experimental evaluation (absolutely necessary)**
- The paper lacks clarity on the presentation of experimental results, especially the absence of confidence intervals despite claims of multiple runs.
- Details on the Office dataset are notably missing.
- It is unclear which experiments address covariate shift. If none, this issue should be discussed in the context of meta-learning, and benchmarks such as MAML or EWC should be possibly evaluated.
- Inconsistencies in the appendix data for the CIFAR dataset and the absence of any details for the Office dataset even in the appendix raise concerns about data accuracy.
- How exactly was the hyperparameter grid search performed? I checked Appendix A.2.2 and didn’t find any description of what metric was used for the grid search to select the best model. How exactly was the best hyperparameter set selected? Was a similar hyperparameter search performed for the benchmarks?
- Ambiguity surrounds the lambda values reported in Figure 3, particularly whether they represent initial values or those after training.

**Strengths And Weaknesses:**

**Strengths**
- The authors address an intriguing aspect of transfer learning, focusing on scenarios where pre-trained models may not have been trained on closely related tasks. This approach is particularly noteworthy as it deals with label shifts, a less commonly explored problem compared to the more typical covariate shifts.
- The methodology proposed is straightforward and appears scalable, making it well-suited for typical transfer learning applications. However, if the pool of models is really large, KNN part of the method could pose a limitation.

**Weaknesses**
- The paper suffers from poor and convoluted writing. There is an inconsistent use of terminology, with the authors often replacing standard terms with their own definitions without clear justification. This makes the paper difficult to read and understand. The paper has a lot of strong statements, which are either wrong or not supported by any evidence.
- The benchmarks used to evaluate the method are limited and do not include the standard benchmarks typically seen in transfer learning research. This omission makes it challenging to accurately assess the effectiveness of the proposed method.

---

> ### Author Response · Authors · 2024-07-28
>
> We are grateful for the reviewer’s suggestions, they are very helpful in improving our work. We will address them one-by-one below and made changes in the revision (marked in red).
>
> - **W1**: We merged a few terms for clarity.
>
>   - “pretrained model”, “source model”, “pre-trained networks” to “source model” in the third paragraph in Introduction.
>
>   - “feature encoder”, “feature extractor”, “pre-logit layer” to “feature extractor” in the whole paper.
>
>   We are happy to make further changes if the reviewer noticed any other inconsistent use of terminology. Moreover, we changed the wording in the revision to make the statements mentioned in “requested changes” less strong.
>
> - **W2**: We added another classic transfer learning benchmark OfficeHome in Appendix A.6.7. The results can also be found in Requested Changes 2.

---

> ### Author Response · Authors · 2024-07-28
> **Requested Changes (Fact-checking and literature research)**
>
> - **RC1**: To address the mentioned issue, we replaced the related sentence in the abstract to the following: “as many existing transfer learning methods typically rely on access to source data, which limits their direct applicability to scenarios where source data is unavailable.” We also cited the mentioned literature (Rame, ICLR 2023) in the related work section paragraph Averaging Model Weights to explain the similarities and differences between our works.
> - **RC2**: To address this concern, we added additional results on Office-Home in Table 13 in the revision. We also added extra details about the dataset in Appendix A.6.7. Our model achieves the highest performance in most cases.
> |  | A,C,R->P  | A,C,P->R  | C,R,P->A  | A,R,P->C |AVG |
> | - | :-:  | :-: |  :-: | :-: | :-: |
> | Model stacking  | 56.77  | 36.67  | 22.56  | 48.38 | 41.10  |
> | Model soup  | 15.48  | 12.80  | 10.63  |14.91  |13.46  |
> | Multi-src SVM  | 69.22  | 42.85  | 24.68  |56.72 |48.37  |
> | MCW  | 68.42  | 43.74  | 22.37  |52.37  |46.73 |
> | DATE   | 59.34  | 38.43  | 21.47  | 45.43 | 41.17 |
> | DECISION  | 53.87  | 37.21 | 21.86  | 42.51 | 38.86  |
> | Independent   | 71.69  | 46.22 | 25.82  | 57.21 | 50.24  |
> | Finetune source   | 71.78  | 45.08  | 26.23  | **59.95**  |50.76 |
> | m=1 | 72.58  | 46.45  | 25.82  |58.35  |50.80 |
> | m=3  | **73.03**  | **47.44**  | **29.51**  | 58.81  | **52.20**  |
> - **RC3**: We apologize for not making it obvious in the paper. Continual learning scenario is mentioned **not** for explaining the use of distance correlation loss **but for** the module-mixing model design. Below we provide an example to explain the benefits of the module-mixing design over direct finetune.
>
>   If Model A is picked to be directly finetuned to create Model B (for task B), then Model B is saved in the repository. Next time, Model B is picked to be directly finetuned to create Model C (task C) ... so on and so forth. Technically, in the described scenario, Model A has been finetuned continuously. And [1] stresses that keep directly finetune causes the loss of plasticity.
>
>   Furthermore, we **explain the motivations** for using dc loss below. (changes are also made accordingly in the revision Section 4.3 paragraph Distance Correlation Loss)
>   - Dc loss encourages features extracted from previous models to be independent from the features learned from the new tasks
>   - Since the output feature dimensions for new and old tasks might not be the same, we need the loss to be able to handle features of different dimensions.
>
>   [1] Continual backprop: Stochastic gradient descent with persistent randomness
> - **RC4**: We provide KNN classification score for quantitative results, and only use UMAP plots for visualization purposes. Since the KNN scores are all relatively high and close for the three different methods, we run the experiment for three times and report the average and standard deviation. The results are added to the revision in Appendix A.6.4. Our model achieves the highest average KNN scores on the testing feature set, thus showed the quality of learned features from the quantitative perspective.
> |  | independent   | finetune   | ours   |
> | - | :-:  | :-: |  :-: |
> | webcam  | 0.975+-0.010  | 0.971+-0.012  | **0.983**+-0.012  |
> | amazon   | 0.920+-0.006  | 0.909+-0.012 | **0.940**+-0.017  |
> | dslr  | 0.961+-0.027  | 0.954+-0.009 | **0.967**+-0.009 |
> - **RC5**: We also provided loss landscape plots as another way of analyzing generalization. These UMAP plots are another indication. We changed the wording in the revision for explaining the UMAP results in Section 5.4.1. We also show the change below.
>
>   “Besides the loss weight landscape analysis, this observation can be seen as another indication that our model has better generalization.”

---

> > ### Comment · Reviewer_uPEN · 2024-08-23
> > **Results on OfficeHome**
> >
> > Could the authors please explain why their results are so much worse than state-of-the-art on OfficeHome? https://github.com/facebookresearch/DomainBed

---

> > > ### Author Response · Authors · 2024-08-26
> > >
> > > Thank you for the valuable feedback and we are happy to explain further.
> > >
> > > There are two main causes for the differences in accuracy. One reason is due to the choice of backbone models. In DomainBed, they use ResNet50, while in the results we show, we used ResNet18 with EFT modules, to be consistent with other experiments in the paper (trainable parameters are 3.9% of the ResNet18 model). The other reason, arguably most important, is that DomainBed studied a very different problem setting from ours, which is called **domain generalization**, an extension of supervised learning where **training datasets from multiple domains (or environments) are available at the same time to train** the predictor. In that sense, the **goal** of domain generalization is **out-of-distribution generalization**: learning a predictor able to perform well in some unseen test domain. In comparison, we **do not have several domains' datasets at the same time** to train one model for better generalization. Instead, we only have access to trained single-domain models, which we can either use as fixed encoders or fine tune using adapters (i.e., EFT modules). This key distinction will be added to the revision in the Related Work Section.

---

> ### Author Response · Authors · 2024-07-28
> **Requested Changes (Proper experimental evaluation)**
>
> - **RC6**: We apologize for the oversight of not providing the standard deviations for CIFAR dataset. We added them to the revised version in Table 10 and 11 in the Appendix.
>
>   To avoid confusion, it is worth noting that the reason why we only ran the experiment on cifar100 multiple times is because of its experimental settings. Specifically, there are 40 tasks in total. For each run, we randomly selected 10 tasks as our starting source model pool. In contrast, the experiments on other datasets are different in that their initial source model pools are the same.
>
> - **RC7**: We added an extra paragraph in Appendix A.2.3 in the revision for detailed explanation. We also show the paragraph below.
>
>   "For each domain in Office31, we randomly select 80% of its data as training set, 10\% as validation set, and 10% as test set.  Data augmentation is performed by using random crops and random horizontal flips. All images have been rescaled to $256 \times 256$ and then cropped to $224 \times 224$. "
>
>  - **RC8**: Office31 addresses the covariate shift problem. Furthermore, we provide results on OfficeHome in Appendix A.6.7 in the revision. We add another sentence in the revision to stress this in Section 5.1.
>
> - **RC9**: To address your concerns, we have added a detailed description of the Office dataset in Appendix A.2.3 in the revision and added the standard deviations to results on CIFAR in Table 10 and 11 in the Appendix. We are happy to make further changes if the reviewer noticed any other inconsistencies in the appendix data.
>
> - **RC10**: For hyperparameter grid search, two learning rates {$10^{−2}$ , $10^{−3}$} , 3 weight decay strengths {0, $10^{−5}$, $10^{−4}$}  and 11 weight initializations {0.001, 0.1, 0.2, 0.3, 0.4, 0.5, 0.6, 0.7, 0.8, 0.9, 0.997} are considered (0.001 and 0.997 are used to avoid setting weights to zero). For each $\lambda_{new}$, other mixing weights in the same layer are set to a same value $(1-\lambda_{new})/m$, with $m$ being the number of selected source models. We select the hyperparameter combination that produces the best validation accuracy.
>
>   We **changed the hyperparameter grid search details paragraph** in Appendix A.2.2 to the paragraph above. This grid search for learning rate, weight decay and weight initialization are **unique to CtrL** dataset in the reported results since CtrL includes a mixture of tasks from different types of datasets, and because the data sizes are very small, which makes the grid search not very time consuming.
>
>   The hyperparameter grid search for learning rate and weight decay **for other datasets** is performed in our **preliminary** results, and the best hyperparameter set are consistent for different domain/tasks from the same dataset. In practice, having a grid search with weight initialization will also help with achieving better performance on new tasks. However, we do not show it since it takes a long time to search for the best weight initialization when the data size is large.
>
> - **RC11**: We added further explanations in the revision for Figure 3 that the lambda values represent their initial values.

---

### Review · Reviewer_kffQ · 2024-07-01

**Summary Of Contributions:**

This paper considers the problem of multi-source data-free supervised transfer learning. Inside the typical setup of transfer learning, the authors investigated the case where the source data are unavailable, and the learner only has access to a group of source models. Depending on whether or not the inside details of source models can be manipulated, the authors further studied two cases: white-box setup and black-box setup. Since there are a large number of source models and not all of them are useful in the target task, this work designs a KNN-based source model selection method to get the most relative $m$ models to the target task. Then, for white-box setup and black-box setup, the authors proposed corresponding solutions to combine the extracted features of the selected source models. Finally, empirical studies validate the effectiveness of the proposed method and the effectiveness of each component of the method via ablation study.

**Audience:**

Yes

**Broader Impact Concerns:**

No broader impact concerns are found.

**Claims And Evidence:**

Yes

**Requested Changes:**

1. Please distinguish between the `\citep` and `\citet` commands. For example, in the second paragraph in the 'Introduction' section, the commands should be `\citep`. I suggest that the authors could have a careful double check of the whole paper for this issue.
2. Please use vector figures such as PDF for pictures, e.g., Figure 1 and results of experiments.
3. There are no hyperlinks for equations, e.g., in the caption of Figure 2.

**Strengths And Weaknesses:**

Strengths:
1. The problem is meaningful since the data of source models are usually unavailable for privacy reasons.
2. The solution is simple, intuitive, and thus easy to understand.
3. The empirical validations are sufficient.

Weaknesses:

My major concern is that the method is mainly heuristic, i.e., without theoretical foundations. I understand that this paper is mainly practical, and requiring corresponding analyses would be beyond the scope. However, without theoretical guarantees, the current technical contributions would be insufficient since, from my point of view, the solution is pretty simple and intuitive. Is it possible that the authors could provide some theoretical analysis for the proposed method?

The performance of the selection method based on KNN is unclear since whether the performance is good or not is large based on the hyper-parameter of $k$ in the kNN method. The authors have studied the impact of $k$ in experiments, but the empirical studies of it are not sufficient to serve as a guide on how to choose such a parameter before the task starts.

When mixing the selected source models in the black-box setup, the authors said that the "target task usually has a relatively small dataset", making me confused. Could the authors provide some further explanations on this point?

---

> ### Author Response · Authors · 2024-07-26
>
> Thank you for your time and careful review. For your **Requested Changes**, we made corresponding edits in the revision.
>
> As for your questions in Weaknesses:
>
> > ... Is it possible that the authors could provide some theoretical analysis for the proposed method?
>
> Unfortunately, we do not have a theoretical analysis for the proposed method. To compensate for the lack of theoretical analysis, we provided extensive empirical analysis to study the model’s properties.
>
> Moreover, we believe that the simplicity of the model design gives us *more flexibility* for adapting models in both white-box and black-box scenarios.
>
> > The performance of the selection method based on KNN ... but the empirical studies of it are not sufficient ...
>
> Although the study of hyperparameter k is empirical, we notice that in general, a larger k provides a better performance, and setting k=5 is enough to produce good performance for most cases tested in the paper.
>
> > ... "target task usually has a relatively small dataset", making me confused...provide some further explanations...
>
> "Target task usually has a relatively small dataset", in this sentence, relatively small dataset means when we compare it to the dataset used to train the large API model. When we deploy the APIs to our own use, we usually have a smaller dataset than the one used to train the API models. Correspondingly, we use a smaller architecture compared to the API for the target model to avoid overfitting.

---

### Review · Reviewer_KKuD · 2024-07-17

**Summary Of Contributions:**

This paper studies the data-free transfer learning, for which there is only access to pre-trained models instead of data for original models. This problem is challenging. Authors propose a model recycling framework for parameter-efficient training of models that identifies subsets of related source models to reuse in both white-box and black-box settings, creating an opportunity for multi-source data-free supervised transfer learning. Pre-trained models are available in white-box scenario, but only the extracted features before the classification head are accessible in black-box scenario. Finally, they perform extensive experiments to highlight the properties of their framework.

**Audience:**

Yes

**Claims And Evidence:**

Yes

**Requested Changes:**

I have some questions, and these could be considered for improve the paper.
1. In Section 4.1, what do you mean about "$t_i = 1$ for the corresponding class in one-hot encoding"? Why this is cross-entropy loss
2. In Algorithm 1, what are $A(f_{s_n};x)$ and $B(f_{t};x)$? How to calculate them?
3. In Table 1, what are the meaning of $D \to W$, $W \to D$, $w / L{DC}$ and $w /o  \ L{DC}$?
4. In the Introduction, show some concrete experimental results to support your conclusions but not just say the conclusion.
5. In Table 5, some parts are just shown as "Task2, Task1 ...", what are they meaning?

**Strengths And Weaknesses:**

##### Strengths
1. They study a challenging problem for data-free transfer learning, and propose a framework for both white-box and black-box setting.
2. Most of the experimental results works well compared with baselines.

##### Weakness
1. In Table 2, this algorithm performs not well.
2. In Section 4.2, how to train $f_{s_n}$ and the weights?

---

> ### Author Response · Authors · 2024-07-26
>
> Thank you for your review and questions, and we are happy to answer them one-by-one below.
>
> - **W1**: As mentioned in the paper, this is indeed a challenging case. However, with this experiment, we intend to show how to train the module-mixing model when facing limited data and when the model pool contains very different types of tasks. Furthermore, we intend to show that the task selection module performs well when the tasks are mixed and are from different sources.
>
> - **W2**: The source models $f_{s_n}$ are trained by minimizing cross-entropy loss as explained in Section 4.2. We freeze the backbone and only train the task-specific modules. We also gave further training details, such as the learning rate and batch size in Appendix A.2. We also stressed this in Section 4.1 in the revision in case the readers share the same confusion.
>
>
> - **RC1**: For $\alpha$-class (multi-class) classification, the log loss for each class is calculated separately and then summed to determine the total loss. $h_{s_n}$ is a $\alpha$-way classifier, while $\mathbf{t}$ is sample $x$’s one-hot label, represented in a vector of $\alpha$ value, with $i$ being the class index. For example, if there are five classes, label 0 in one-hot coding is [1, 0, 0, 0, 0]. We added extra explanations in the revision in Section 4.1.
>
> - **RC2**: $A(f_{s_n}; x)$ and $B(f_{t}; x)$ are distance matrices first mentioned in equation 5. Please see the definitions below equation 5 in the paper for how to calculate them. We also made further notification in Algorithm 1 in the revision.
>
> - **RC3**: $A, D \rightarrow W$ means, $A, D$ are the source domains, while $W$ is the target domain; Similarly, $A, W \rightarrow D$ means, $A, W$ are the source domains, while $D$ is the target domain. $w/ L_{DC}$ stands for training module-mixing model with distance correlation loss, while $w/o L_{DC}$ stands for training module-mixing model without distance correlation loss. This is meant as an ablation study to learn the effects of DC loss.
>
> - **RC4**: We discussed some of the experimental results in the last paragraph of the introduction in the revision.
>
> - **RC5**: They represent “models trained for Task2, Task1...” that our source model selection module selected. For instance, when $m=2$, our source model selection module selects the models trained for Task2 and Task 3 when training target task Task1. The selected tasks’ names are colored to provide more clarity. In this example, Task2 is in blue and Task3 is in green as shown in the row header. We also stressed this in Table 5 and in the experiment section in the revision.

---

### Author Response · Authors · 2024-07-28
**To all Reviewers**

Dear Reviewers,

We thank you kindly for your effort and time in reviewing our paper. In response to the requested changes, we provide the revision on openreview, which we referenced many times in the replies below. Please let me know if we could further address your questions, and we will be happy to answer them.

Thank you!

---

> ### Comment · Action_Editor_GWWp · 2024-07-28
> **thank you**
>
> Hi authors,
>
> Based on the TMLR process
>
> "Authors may post rebuttals and update their papers in response to the reviews, and the reviewers and editor may privately discuss the paper."
>
> you should be able to just revise your paper on TMLR instead of posting on 3rd-party platforms. Can you do so?
>
> Thanks.
>
> Your AE

---

> ### Author Response · Authors · 2024-07-29
>
> Thank you for the reminder! I just uploaded the revision to openreview and removed the one on 3rd-party platform.

---

### Decision · Action_Editor_GWWp · 2024-09-16

**Recommendation:** Reject

**Comment:**

This paper focuses on source-free transfer learning, where only pre-trained models are accessible, and original source data is unavailable due to privacy concerns. The authors propose a model recycling framework that identifies and reuses subsets of pre-trained models in both white-box and black-box settings. The framework enables Model as a Service providers to create efficient model libraries for multi-source data-free supervised transfer learning. The proposed solution is simple, which should be appraised. The authors demonstrate their method using various experiments, but reviewers express concerns about the strength of the overstated claims, especially in terms of positioning in concurrent transfer learning literature. In addition, reviewers share concerns about lack of theoretical grounding and limited experimental results (such as the proof-of-concept with small unrealistic cases) to support the claims. The authors are also encouraged to improve the clarity on presenting the experimental results.

**Audience:**

yes

**Claims And Evidence:**

not fully---lack of theoretical evidence, literature positioning is questionable, and empirical evidence can be enriched